Fast sequence-based microsatellite genotyping development workflow

Lepais Olivier olivier.lepais@inra.fr olivier.lepais@inrae.fr 1 2
Chancerel Emilie 1
Boury Christophe 1
Salin Franck 1
Manicki Aurélie 2
Taillebois Laura 2
Dutech Cyril 1
Aissi Abdeldjalil 3
Bacles Cecile F.E. 2
Daverat Françoise 4
Launey Sophie 5
Guichoux Erwan 1
1 INRAE, Univ. Bordeaux, BIOGECO , Cestas , France
2 INRAE, Université de Pau et Pays de l’Adour, ECOBIOP , Saint-Peé-sur-Nivelle , France
3 LAPAPEZA, University of Batna 1 Hadj Lakhdar , Batna , Algeria
4 INRAE, EABX , Cestas , France
5 INRAE, Agrocampus Ouest, ESE, Ecology and Ecosystem Health , Rennes , France
Thomas Jonathan
Electronic publication date: 2020 May 4
Publication date: 2020
Volume: 8
Electronic Location ID: e9085
Received 2020 Jan 3; Accepted 2020 Apr 8
Copyright: ©2020 Lepais et al.
Copyright year: 2020
Copyright holder: Lepais et al.
License: This is an open access article distributed under the terms of the Creative Commons Attribution License, which permits unrestricted use, distribution, reproduction and adaptation in any medium and for any purpose provided that it is properly attributed. For attribution, the original author(s), title, publication source (PeerJ) and either DOI or URL of the article must be cited.
License URL: https://creativecommons.org/licenses/by/4.0/

Keywords: Sequence-based microsatellite genotyping, SSR-GBS, SSR-seq, Haplotype sequence, HapSTR, SNPSTR

Funding: Agence de l’Eau Adour-Garonne 2017/3273 Région Nouvelle-Aquitaine 2016-1R20602-00007239 Agence Française pour la Biodiversité 2016–18 A13 INRAE Department of Agronomy ISVSA of the University Batna 1 Hadja Lakhdar Genome Transcriptome Facility of Bordeaux Investissements d’Avenir, Convention attributive d’aide EquipEx Xyloforest ANR-10-EQPX-16-01 This research was supported by grants of the Agence de l’Eau Adour-Garonne (project 2017/3273), the Région Nouvelle-Aquitaine (project 2016-1R20602-00007239), the Agence Française pour la Biodiversité (project 2016–18 A13) and INRAE. Abdeldjalil Aissi benefited from a travel grant from the Department of Agronomy ISVSA of the University Batna 1 Hadja Lakhdar. Technical developments and sequencing were performed at the Genome Transcriptome Facility of Bordeaux (Grants from Investissements d’Avenir, Convention attributive d’aide EquipEx Xyloforest ANR-10-EQPX-16-01). There was no additional external funding received for this study. The funders had no role in study design, data collection and analysis, decision to publish, or preparation of the manuscript.

==============================
Application of high-throughput sequencing technologies to microsatellite genotyping (SSRseq) has been shown to remove many of the limitations of electrophoresis-based methods and to refine inference of population genetic diversity and structure. We present here a streamlined SSRseq development workflow that includes microsatellite development, multiplexed marker amplification and sequencing, and automated bioinformatics data analysis. We illustrate its application to five groups of species across phyla (fungi, plant, insect and fish) with different levels of genomic resource availability. We found that relying on previously developed microsatellite assay is not optimal and leads to a resulting low number of reliable locus being genotyped. In contrast, de novo ad hoc primer designs gives highly multiplexed microsatellite assays that can be sequenced to produce high quality genotypes for 20–40 loci. We highlight critical upfront development factors to consider for effective SSRseq setup in a wide range of situations. Sequence analysis accounting for all linked polymorphisms along the sequence quickly generates a powerful multi-allelic haplotype-based genotypic dataset, calling to new theoretical and analytical frameworks to extract more information from multi-nucleotide polymorphism marker systems.

Introduction

In the age of high-throughput sequencing (HTS) technologies, and in comparison with SNP genotyping which is gaining momentum, application of HTS to microsatellite genotyping was until recently lagging behind. Traditional capillary electrophoresis-based microsatellite genotyping suffers from several drawbacks: homoplasy (alleles of identical size having different underlying sequence; Viard et al., 1998; Estoup, Jarne & Cornuet, 2002), time and cost consuming development and genotyping, low throughput, lack of automation and data standardization (Moran et al., 2006; Ellis et al., 2011). Yet, all of these limitations are linked to the fact that currently microsatellite genotyping relies on allele discrimination based on amplicon size assessed by capillary electrophoresis (De Barba et al., 2016) and do not hold true if microsatellite genotyping transitions to sequence-based genotyping. Previous direct comparisons of capillary electrophoresis and sequence-based microsatellite genotyping (called SSRseq thereafter) validated SSRseq as a reliable method (Darby et al., 2016; Vartia et al., 2016). Advantages of sequence-based over capillary electrophoresis-based microsatellite genotyping are significant. Direct access to allele sequence reveals additional polymorphisms that remain hidden when using only allele size to identify variation (Darby et al., 2016; Vartia et al., 2016; Šarhanová et al., 2018). Sequence data thus reduces allele homoplasy because alleles of the same size may contain molecular variation that does not translate into size variation such as SNP, indels masking variation in repeat number, or presence of two adjacent SSR motifs with complementary size variation (Darby et al., 2016). As a result, SSRseq offers refined genetic diversity estimation and population structure inference (Darby et al., 2016; Bradbury et al., 2018; Neophytou et al., 2018; Viruel et al., 2018; Layton et al., 2020).

Updating microsatellite genotyping to modern technologies remains important for several reasons. Firstly, some current scientific questions in ecology or evolutionary biology can be answered using a moderate number (e.g., a dozen (Harrison et al., 2013) to a hundred (Bradbury et al., 2018; Layton et al., 2020)) of highly polymorphic multi-allelic loci such as microsatellites. Secondly, variation in the number of repeated oligonucleotide motif is a unique kind of polymorphism with specific mutation mechanism and rate which in itself provides a complementary picture of genetic variation to nucleotide substitutions across populations (Haasl & Payseur, 2011) and genomes (Willems et al., 2014). Thirdly, it becomes more and more obvious that microsatellite polymorphism is involved in numerous biological processes such as gene expression regulation and epigenetic mechanisms (Bagshaw, 2017; Sadd et al., 2018), and more generally in phenotypic variation (Xie et al., 2019) including human diseases (Gymrek, 2017; Hannan, 2018). Thus, while marker preference evolves through time with specific markers dominating the genotyping field over a period of time following technological advances (Schlötterer, 2004; Seeb et al., 2011), maintaining our capability to interrogate any kind of polymorphism in the context of rapid HTS technological advances is paramount and should be prioritized.

Studies published so far have explored specific technical or analytical aspects of SSRseq. Several bioinformatics approaches have been developed (Hoogenboom et al., 2016; Suez et al., 2016; Zhan et al., 2017; Barbian et al., 2018), different laboratory protocols tested (Vartia et al., 2016; Pimentel et al., 2018) and means to account for molecular variation on population genetic inference compared (Neophytou et al., 2018; Šarhanová et al., 2018; Viruel et al., 2018; Curto et al., 2019). Together, these studies explored numerous issues surrounding the technical and analytical advantages of SSRseq over traditional methods.

Here, we propose an integrative workflow for the development of a SSRseq analysis for application to non-model species. We apply this workflow to five species groups from families taken across phyla (Basidiomycota: Armillaria ostoyae; Angiosperms: Quercus faginea and Q. canariensis; Euarthropoda: Melipona variegatipes; Chordata: Alosa alosa, A. fallax and Salmo salar) that markedly differ in the amount of genomic data already available for them. We compare a broad range of possible development scenarios including sequencing of already optimized microsatellite assay traditionally genotyped on capillary sequencers, optimizing primers around already developed microsatellites, and developing microsatellite de novo from a range of genomic resources when available, or from newly generated low coverage random genome sequences for species without existing genomic resources. Building on our previous experience in development of highly multiplex microsatellite genotyping protocols (Guichoux et al., 2011; Lepais & Bacles, 2011), we propose a streamlined approach with demonstrated application to groups of species with a wide range of genetic and evolutionary characteristics. We applied a microsatellite sequence data analysis pipeline to produce haplotypic data accounting for all polymorphisms detected in sequenced alleles, validated by extensive blind-repeat genotyping to estimate SSRseq error rates. We emphasize that efficient and powerful multi-polymorphism haplotype-based genotyping approaches are easy to develop and apply, calling for new theoretical and analytical development to extract more information from multi-polymorphism haplotypes.

Material and Methods

Studied species, SSRseq development strategies and DNA isolation

We selected a range of species from different Kingdoms among biological models studied in our laboratories (Table 1). For S. salar we took the most straightforward route by amplifying and sequencing microsatellites using previously developed primers. To develop a refined workflow, we chose species with different level of genomic resource availability to test alternative de novo microsatellite development strategies that are likely to cover a wide range of situations (Table 1).

Table 1 SSRseq development strategy and DNA characteristics of species used in this study.

Kingdom	Class	Species	SSRseq development strategy	Number of DNA extracted individuals	DNA extraction protocol	DNA quality	Reference	
Animalia	Actinopterygii	Salmo salar	Previously developed loci	1,152	Salt-chloroform (Gauthey et al., 2015)	High	Bacles et al. (2018) and Lepais et al. (2017)	
Plantae	Eudicots	Quercus faginea, Q.canariensis	Re-designed primers around already developed loci using reference genome sequence of closely related species	380	Invisorb DNA Plant HTS 96 kit	High	This study	
Animalia	Actinopterygii	Alosa alosa, A.fallax	De novo loci development based on available repeat-enriched library sequencing	382	Invitrogen PureLink Genomic DNA Mini kit	Highly degraded	Taillebois et al. (2020)	
Fungi	Agaricomycetes	Armillaria ostoyae	De novo loci development based on reference genome sequence	384	CTAB (Prospero, Lung-Escarmant & Dutech, 2008)	Heterogeneous	This study	
Animalia	Insecta	Melipona variegatipes	De novo loci development based on newly generated low coverage whole genome shotgun sequencing	91	Qiagen DNeasy 96 Blood & Tissue Kit	High	This study	

SSRseq using previously-developed primers

The most straightforward approach to SSRseq microsatellite genotyping, i.e., based on sequence information from existing primers, was applied to S. salar using two marker selection strategies. In the first strategy, we selected 23 primers from a list of 81 microsatellites available for S. salar (O’Reilly & Wright, 1995; Slettan, Olsaker & Lie, 1996; Ozaki et al., 2001; Rexroad et al., 2001; Gilbey et al., 2004; Paterson et al., 2004; King, Eackles & Letcher, 2005; Vasemägi, Nilsson & Primmer, 2005; Thorsen et al., 2005; Yano et al., 2013, see details in Table S1). Selection criteria included allele size smaller than 300 bp to ensure that sequencing reads can span the entire allele length including library construction and absence of sequences complementarity between primers tested using Multiplex Manager (Holleley & Geerts, 2009). In the second strategy, we chose to sequence a set of 15 microsatellites (Table S1) that are routinely amplified in a single multiplexed PCR and genotyped using standard capillary electrophoresis (Bacles et al., 2018; Lepais et al., 2017).

SSRseq with microsatellite (re)development

Genomic resources for microsatellite (re)development

For the other species, microsatellites primers were either re-designed or developed de novo from various genomic resources (Table 1, Fig. 1 top panel).

Figure 1 Workflow for SSRseq markers optimization or development depending on genomic resource availability, from selection to multiplexed amplification and library preparation to bioinformatics analysis.

For Quercus sp., primers were re-design primers in flanking regions of existing microsatellite markers to optimize multiplex amplification and sequence interpretability while taking advantage of already validated microsatellite markers. We extracted primer sequences from 259 polymorphic and mapped EST-derived (Durand et al., 2010) and 35 genomic microsatellites (Steinkellner et al., 1997; Kampfer et al., 1998). The primer sequences were mapped on the Q. robur reference genome (Plomion et al., 2018, GenBank accession GCA_003013145.1) using bowtie 2 v2.3.4.1 (Langmead & Salzberg, 2012) and genomic sequence spanning from 200 bp downstream of the forward primer to 200 bp upstream of the reverse primer position were extracted using bedtools v2.25.0 (Quinlan, 2014) resulting in 294 sequences used as genomic resource to design new primers.

For Alosa sp., we used sequences obtained from a Roche 454 GS-FLX sequencing run on a microsatellite-enriched DNA library (Rougemont et al., 2015) following the method described in Malausa et al. (2011).

For A. ostoyae, we used the reference genome sequence as genomic resource to identify microsatellites loci (Sipos et al., 2017, GenBank accession GCA_900157425.1).

Finally, no genomic resources were available for M. variegatipes. We therefore used DNA from one individual to construct a whole-genome sequencing library using Illumina TruSeq DNA kit. The resulting library was sequenced on an Illumina MiSeq flowcell using v3 2x300 pb paired-end sequencing kit. Mothur software v1.39.5 (Schloss et al., 2009) was used to assemble paired reads and keep paired reads with a minimum overlap of 100 bp without mismatch. We randomly subsampled 500,000 reads from the resulting 6.74 million paired reads for subsequent microsatellite identification, because a few hundreds of thousands of random sequences are sufficient to identify thousands of microsatellites (Castoe et al., 2010; Lepais & Bacles, 2011; Curto et al., 2019).

Table 2 Summary of the tested scenarios for SSRseq genotyping.

Species	SSRseq development strategy	Candidate loci	Screened loci	Number of loci in a single multiplexed PCR	Sequencing Plateform	Total number of individuals sequenced	Number of individual analyzed for this study	Number of repeated individuals	Sequenced locia	Mean (cv)sequences/ loci/ individual	Reliable loci genotypedb	Overall success rate	
S. salar	Previously developed loci	81	–	23	Ion Torrent PGM i316	960	66	66	20	161 (61%)	9	39%	
				23	Illumina MiSeq - 1/2 nano PE	192	96	96	20	70 (41%)	10	43%	
S. salar	Previously developed loci	15c	–	15	Illumina MiSeq - 1/2 nano PE	192	96	96	13	99 (65%)	7	47%	
Quercus sp.	Primer redesign around previously developped loci	462	60	60	Illumina MiSeq - 1/3 V2 PE	380	46	46	53	260 (32%)	40	67%	
Alosa sp.	De novo	2,872	60	28	Illumina MiSeq - 1 nano SE	382	156	156	25	95 (58%)	21	75%	
				28	Illumina MiSeq - 2 nano SE	382	156	156	26	198 (58%)	24	86%	
				28	Illumina MiSeq - 3 nano SE	382	156	156	26	267 (58%)	24	86%	
A. ostoyae	De novo	1,806	60	51	Illumina MiSeq - 1/2 V2 PE	384	384	96	48	243 (83%)	38	75%	
M. variegatipes	De novo	8,937	60	54	Illumina MiSeq - 1/4 V2 PE	182	91	91	49	176 (45%)	39	72%	
Notes.

a Loci showing substantial evidence for minimum sequencing success (at least 20 sequences in at least 50% of the individuals).

b Reliable loci (less than 50% of missing data among individuals and less than 6% of genotyping error based on comparison of repeated genotyping).

c Routinely genotyped using optimized multiplexed PCR and capillary-based sequencer. FDSTools analysis using two parameter sets: stutterfinder -s:-1:50, +1:10 allelefinder -m 15 -n 20; and stuttermark -s:-1:70, +1:10 allelefinder -m 10 -n 20. For each marker, four parameter combination were used (two strategies and two parameters set) and for each strategy, the best parameter set was used for a given locus.

de novo microsatellite development or primer re-design

The command line version of QDD pipeline v 3.1 (Meglécz et al., 2010; Meglécz et al., 2014) was run on either (i) a reference genome sequence, (ii) a set of low coverage random sequences or (iii) sequence extracted around already characterised microsatellite loci (Table 2) , to detect sequences containing microsatellites, identify good quality sequence (singletons and consensus) from problematic sequences (sequences showing low complexity, minisatellites or multiple BLAST hits with other sequences) and design primer pairs flanking the identified microsatellites (Fig. 1). QDD pipeline was run with default parameters, except for the primer design step (pipe3) where parameters were stringently defined in order to improve their capacity to be amplified jointly in a single multiplexed PCR (Qiagen Multiplex kit handbook; (Lepais & Bacles, 2011)): primer optimal size was set to 25 nucleotides (min: 21, max: 26), optimal annealing temperature to 68 °C (min: 60 °C, max: 75 °C) with a maximal difference of 10 °C between primers of a same pair and optimal percentage of cytosine and guanine of 50% (min: 40%, max: 60%). In addition, PCR product size was set between 120 and 200 bp to be compatible with a wide range of sequencing platforms and to produce robust genotyping assays that can be used to analyse degraded or low quantity DNA samples. QDD analysis results in a large number of candidate loci with designed primer pairs from which a restricted number of loci can be selected (Fig. 1). At the exception of Quercus sp. where a restricted set of input sequences necessarily limited choice among resulting candidate microsatellites, several quality criteria were used to select 60 microsatellites among hundreds to thousands candidates for further testing. We followed recommendations of Meglécz et al. (2014) to prioritize primer pairs with increased likelihood of amplification success by selecting microsatellite from singletons and not from consensus sequence, with pure repeat instead of compound motifs, showing at least 20 bp between the primers and the repeat motif, and with flanking region showing high complexity (e.g., primer pairs from the design group A following QDD terminology: no minisatellite, no other microsatellite in the flanking region, no homopolymer in the flanking region or the primer). In addition, we further selected microsatellites with the highest number of repeats to increase the probability of selecting polymorphic loci, avoided motif that can form hairpin such as AT repeats and when possible included a variety of di-, tri- and tetra nucleotide repeats.

Primer modification and simplex amplification tests

For Ion Torrent sequencing (Table 2), tag A 5′-GCCTTGCCAGCCCGCTCAG- 3′ was add to the 5′ end of each forward primer and tag B sequence 5′-GCCTCCCTCGCGCCA- 3′ (Margulies et al., 2005; Blacket et al., 2012) was added to the 5′ end of each reverse primer. For Illumina sequencing (Table 2), specific tags 5′-TCGTCGGCAGCGTCAGATGTGTAT AAGAGACAG- 3′ and 5′-GTCTCGTGGGCTCGGAGATGTGTATAAGAGACAG- 3′ were added to the 5′ end of the forward and reverse primer sequences respectively. Designed primers were tested for potential primer dimer formation using Primer Pooler (Brown et al., 2017). Primer pairs showing a deltaG lower than −6 kcal/mol are likely to form dimer and result in poor amplification in a multiplexed PCR. For locus involved in significant interactions, alternative primers were selected or in absence of alternative, another locus was selected from the candidate list. Oligonucleotides were ordered in a plate format from Integrated DNA Technologies with standard desalt purification at a final concentration of 100 µM. Primer pairs were tested using simplex amplification of one individuals per species using Qiagen Multiplex kit in a final volume of 10 µL and with a final concentration of each primer of 0.2 µM. Amplification conditions consisted of an initial denaturation step at 95 °C for 15 min, followed by 35 cycles consisting of denaturation at 95 °C for 20 s, annealing at 59 °C for 60 s and extension at 72 °C for 30 s, and a final extension step for 10 min at 72 °C. Amplicons and 1 Kb size standard were then loaded on a 3% agarose gel containing GelRed or SyberSafe dye and migrated at 100 v for 15 min. Each locus was screened under UV light for positive amplification with a clear band at the expected size.

Multiplex microsatellite amplification and sequencing library construction

From 192 to 960 individuals were analyzed depending on the taxa considered including from 46 to 156 repeated individuals to check the reproducibility of the method (Table 2). For each of the taxonomic groups, a three-round multiplex PCR approach was used to amplify all loci simultaneously and improve amplification homogeneity and thus coverage of sequence between loci (Chen et al., 2016). In the first round, a multiplexed PCR including all selected locus primers was performed (Fig. 1, Table S1 for locus characteristics including primer sequences). PCR amplification were carried out in 96-well plates in a final volume of 5 µL or 10 µL using Qiagen Multiplex kit, 0.05 µM of each forward and reverse tailed primers and about 40 ng of template DNA (depending on the species, 1 µL of undiluted or diluted isolated DNA). PCR cycles were performed on Applied Biosystems 2720 or Verity thermocyclers and consisted of a denaturing step of 5 min at 95 °C followed by 20 cycles of 95 °C for 30 s, 59 °C for 180 s and 72 °C for 30 s (Qiagen Multiplex kit handbook; Lepais & Bacles, 2011). In the second round, additional Taq polymerase added with the aim to use remaining primers completely. The PCR mixture of a final volume of 5 or 10 µL consisted of 2.5 or 5 µL of Qiagen Multiplex kit, 1.5 or 3 µL of undiluted amplicon and 1 or 2 µL of water. The PCR cycles were identical as in the first round. The third round is the indexing PCR (Fig. 1) that add Ion Torrent or Illumina sequencing adaptors and barcodes used to assign each sequence to an individual. For Ion Torrent sequencing, we used 106 different barcodes resulting in a total of 960 barcode combinations. For Illumina sequencing, we used the Nextera XT index set allowing for 384 barcode combinations. The PCR mixture of a final volume of 10 µL consisted of Qiagen Multiplex kit Master Mix, 0.5 µM of sequencing platform-specific adaptor and 5 µL of undiluted amplicon resulting from the second PCR round. PCR cycles consisted of a denaturing step of 5 min at 95 °C followed by 15 cycles of 95 °C for 30 s, 59 °C for 90 s and 72 °C for 30 s and a final extension step of 68 °C for 10 min. Amplicons from the 96 wells within a plate were pooled together in an Eppendorf tube, and purified with 1.8X Agencourt AMPure XP beads (Beckman Coulter, UK). Quality check and quantification were done using Agilent Tapestation D1000 kit and Qubit fluorometric system (Thermo Fisher Scientific), and quantified using Kapa libraryquantification kit in a Roche LightCycler 480 quantitative PCR. The resulting two to ten pools were pooled in equimolar concentration and sequenced using an Ion Torrent PGM i316 chip or Illumina MiSeq flowcell using nano or v2 2 × 250 bp paired-end sequencing kit (Table 2) at the Genome Transcriptome Facility of Bordeaux.

Bioinformatics data analysis

Sequence preparation

After sequence demultiplexing and adaptor trimming using a sequencer platform built-in software, quality was controlled using FastQC (http://www.bioinformatics.babraham.ac.uk/projects/fastqc/) and reads shorter than 70 bp were removed using cutadapt (Martin, 2011). When paired-end sequencing was used, paired reads were assembled into contigs using pear (Zhang et al., 2014) with the default scoring method based on assembly score (allowing for mismatch and accounting for base quality scores), a minimum overlap of 50 bp and a maximum assembled sequence length of 450 bp. For Alosa sp., reverse reads quality was generally poor, therefore, only the forward read was used as its length (250 bp) encompasses the whole length of the sequenced loci (max. 200 bp). In some cases, microsatellite amplicons from several species were pooled prior to PCR indexing so that two individuals from two species shared an identical barcode combination and were sequenced in a single run. Forward primer sequences of species-specific loci were used to sort sequences belonging to different species into different fastq files using fqgrep tool (https://github.com/indraniel/fqgrep) allowing for one mismatch.

Converting microsatellite sequences to genotypes

We used FDSTools v1.1.1 pipeline (Hoogenboom et al., 2016) to identify sequences corresponding to the microsatellite alleles and call genotypes for each individual (Fig. 1). This analytical tool was chosen because it accounts for any kind of polymorphism detected across the analysed sequences (including variation in the number of repeated motifs, SNP or indels) while integrating specific tools to detect true allele from stutter mutation introduced during amplification that are typical of microsatellite markers.

First, tssv (Anvar et al., 2014) matches primer sequences, allowing for 8% of mismatch, to identify sequences originating from each locus and count the occurrence of each unique sequence found for each locus for each individual (Fig. 1). Then, Stuttermark uses the number of repeats of the microsatellite motif and the coverage of each unique sequence to flag unique sequences as potential allele, stutter resulting from slippage mutation during PCR and erroneous sequences (Fig. 1). Finally, Allelefinder calls one or two alleles among the most abundant sequences flagged as potential alleles by Stuttermark (Fig. 1). Following the FDSTools analysis, several custom-made bash routines were used to format the tabulated genotypic table, compare genotypes from repeated individuals to estimate locus specific allelic error rate defined by the number of allele mismatches between replicated genotypes divided by the number of alleles compared. In addition, allele level information was extracted including allele sequences, three-digit code used for genotype annotation, number of occurrence across individuals and allele length. Locus characteristics such as missing data rate and number of alleles are also summarized across the analysed individuals (Fig. 1). All these bioinformatics steps have been embedded into a single bash script (SSRseq_DataAnalysis_ParametersComparison.sh available at https://doi.org/10.15454/HBXKVA) that allow to modify key analytical parameters (Supplemental Information S1) to evaluate their effect on the quality of the genotypic call (number of detected alleles, error and missing data rates).

Two analytical strategies were compared. In the first strategy (called FullLength thereafter), all variation identified between primers was considered as a haplotype irrespectively of the nature of the polymorphism because all polymorphisms are physically-linked to each other in reads that encompass the whole locus. The FullLength strategy may be too complex for some loci or species showing high levels of polymorphism. In the second strategy, the analysis was therefore restricted to the repeat motif only (strategy called RepeatFocused thereafter). In this case, primer and flanking sequences surrounding the repeated motif are indicated in the primer sequence field of the FDSTools input file. The tssv step then extracts the sequence corresponding to the repeat motif region (still allowing for 8% of mismatch which will accommodate flanking sequence polymorphism) to perform subsequent genotypic call with Stuttermark and Allelefinder as described above. As the analytical approach used by FDSTools is based on counting coverage of unique sequences, any variation identified within the repeated motif region, including variation in repeated motif number, SNP and indel within the motif region, will still be accounted for when defining alleles. While this RepeatFocused strategy may be more robust due to the shorter length of sequence analysed, it should identify a smaller number of alleles compared to the FullLength strategy.

Comparing analytical approaches

For each locus, we determined the best analytical strategy using the following criteria by order of importance: estimated allelic error, amount of missing genotypes and number of detected alleles. Loci that showed more than 6% of allelic error or more than 50% of missing data across individuals (within each species group) were flagged as failed and removed from further inspection. These arbitrary thresholds have been chosen to remove bad quality loci while conserving moderate quality loci (see Table S1 for locus specific missing data and allelic error rates). Decreasing these thresholds will have resulted in the removal a handful of loci for each species, the majority of loci having low genotyping error and missing data (Table S1). For each species and analytical strategy, we recorded the number of genotyped loci, mean allelic error and missing data rate and the total number of alleles (haplotypes) across loci. We then determined the best overall approach for each locus and used it to genotype each locus generating a final Combined genotypic dataset for each species using a specific bash script (SSRseq_DataAnalysis_FinalGenotyping.sh available https://doi.org/10.15454/HBXKVA). Finally, the overall development success rate was computed for each species by dividing the number of reliable loci by the number of loci included in the multiplexed PCR.

Gain from sequence information

The number of identified alleles based on sequence information (haplotypes) was compared to the number of alleles differing in amplicon length only for all analysed loci to assess the gain of information obtained by using sequence data and estimate size homoplasy. We investigated further the nature of the detected polymorphism by counting, for each locus, the number of variations in the number of repeats, SNP and indels in the repeated motif and the flanking regions that differ between haplotypes.

Results

Sequence-based genotyping of previously developed microsatellites

Attempts to genotype previously developed microsatellites in S. salar either from a new combination of 23 loci or a routinely-used multiplex of 15 loci resulted in a low number of reliable loci genotyped (Table 2). The overall success rate (the percentage of reliable loci over the number of loci amplified in the multiplexed PCR) ranged from 39% to 47% and the number of reliable loci from 7 to 10 (Table 2). Moreover, the quality of the generated genotypic datasets is relatively low with a rate of missing data and allelic error above 10% and 1% respectively (Fig. 2). The sequences produced on the PGM Ion Torrent platform resulted in the lowest genotypic data quality (Fig. 2). The same genotyping protocol sequenced on the Illumina MiSeq platform produced higher genotypic quality with lower missing data and allelic error rates (Fig. 2) in spite of a 2.3 times lower mean coverage per sequenced locus per individual (Table 2). The lowest performance of the Ion Torrent platform is due to the higher sequencing error rate linked to spurious insertion-deletion around homopolymer tracts. This results in a waste of sequencing reads, increasing noise (e.g., erroneous singletons: unique sequence with a coverage of one or a very few reads), at the expense of sequence exactly matching the true alleles. The Illumina MiSeq sequencing platform was thus used for subsequent analyses in other species.

Figure 2 Results of SSRsq development from previously developed microsatellites.

S. salar for (A) a new multiplex of 23 microsatellite sequenced with Ion Torrent PGM and (B) Illumina MiSeq sequencing platforms, and (C) a routinely-used multiplex of 15 microsatellites sequenced with Illumina MiSeq sequencing platform. Number of reliable loci, total number of alleles, missing data and allelic error rates are indicated for three bioinformatics analysis strategies that focused either on all polymorphism across the sequence, on polymorphism within the repeated motif only, or a combination of the best strategy for each locus.

Sequence-based genotyping of de novo developed microsatellites

In contrast, overall success rate of de novo microsatellite development ranged from 67% to 86% (Table 2). Given the high number of candidate microsatellites typically identified from high-throughput sequencing or reference genome sequence, we were able to screen as much as 60 new loci, and combined most of them (from 28 to 60) in a single multiplexed PCR for amplification (Table 2). As a result, the final number of reliable loci was consistently high amounting to 24 for Alosa sp., 38 for A. ostoyae, 39 for M. variegatipes and 40 for Quercus sp. (Table 2). All these protocols produced high quality genotypic dataset, with low missing data (2.6% for Quercus sp., 6.8% for Alosa sp., 5.4% for M. variegatipes) at the exception of A. ostoyae (20.7%) and low allelic error rates as estimated based on blind-repeat genotyping (0.4% for Quercus sp., 0.6% for Alosa sp., 0.9% for M. variegatipes and 0.7% for A. ostoyae). For Alosa sp. and Quercus sp. all reliable loci were found to be transferable between species.

We explored the effect of sequence coverage on genotypic data quality on Alosa sp. by sequencing the same set of 28 microsatellites amplified in 156 individuals using one, two or three Illumina MiSeq nano flowcells (Table 2). The resulting increased coverage, from 95 to 198 and 267 sequences respectively per locus per individual (Table 2), recovers more data for those highly degraded DNA samples. First, increasing the coverage from 95 to 198 sequences by locus by individual detected three additional loci, while a further increase in coverage failed to recover additional locus (Table 2). Second, the missing data rate linearly decreases with the increase in coverage, from 16.4% to 9.0% and 6.8% with 95, 198 and 267 sequences per locus per individual respectively (Fig. S1). It is worth noting that the allelic error rate is not affected by genome coverage, as it varied only slightly between 0.5% and 0.7% without any correlation to coverage (Fig. S1). A significant result is that even in conditions when only highly degraded DNA templates are available, reliable genotypic data can be obtained with moderate coverage, while increasing coverage will reduce missing data but not genotyping error rate.

Whole sequence VS repeated motif polymorphism analysis

Focusing the analysis on the repeated motif slightly increased the number of reliable loci and tends to produce marginally fewer missing data and allelic errors (Fig. 3). However, numerous polymorphisms that may be present in the flanking sequences are not accounted for. Indeed, analysing all polymorphism detected between the PCR primers resulted in a higher mean number of allele per locus, at the expense of slightly higher missing data and allelic error rates (Fig. 3). Interestingly, 17% of the loci can be analysed reliably using either the FullLength or the RepeatFocused analytical approach. Thus combining analytical strategies by selecting the best approach for each locus resulted into an optimized dataset (Fig. 3). Even for loci with reliable genotypes irrespectively of the analytical approach chosen, selecting the one that produces the best quality data (in terms of number of alleles, missing data and error rate) leads to an improved dataset quality. This combined strategy results in recovering the highest number of loci and alleles while keeping missing data and allelic error rates at the lowest (Fig. 3).

Figure 3 Results of SSRseq development based on newly optimized microsatellites.

(A) Quercus sp., (B) Alosa sp., (C) A. ostoyae and (D) M. variegatipes sequenced with Illumina MiSeq sequencing platform. Number of reliable loci, total number of alleles, missing data and allelic error rates are indicated for three bioinformatics analysis strategies that focused either on all polymorphism across the sequence, on polymorphism within the repeated motif only, or a combination of the best strategy for each locus.

Types of polymorphism detected across species

While most common population genetics applications do not necessitate to characterize the nature of the polymorphism differentiating alleles, the main advantage of sequence data (in addition to analysing a much higher number of loci) is to be able to identify allelic variation that does not translate into size variation, i.e., the only variation that is detected when using classical electrophoretic approaches. Across species, the proportion of alleles would have remained undetected by capillary electrophoresis (size homoplasy) ranging from 6% for M. variegatipes, 11% for S. salar, 14% for Alosa sp., 35% for Quercus sp. and 53% for A. ostoyae (Table 3). Conversely, the increase in the proportion of allele detected by accessing sequence data ranges from 6% for M. variegatipes, 13% for S. salar, 16% for Alosa sp., 56% for Quercus sp. and 113% for A. ostoyae (Table 3). Indeed, beside variation in repeat number, we identified numerous SNP and indel either in the flanking sequence or in the repeat motif itself (Fig. 4, Table 3). In fact, additional polymorphism beyond variation in repeat number was the rule rather than the exception (Table 3). However, differences in the proportions of the type of the detected polymorphism were found between species (Fig. 4). While repeat number variation represented more than 80% of the polymorphism detected in S. salar, Quercus sp., Alosa sp. and M. variegatipes, SNP in the flanking sequence was the most frequent polymorphism for A. ostoyae representing 49.8% of the variation, much higher than variation of repeat number estimated at 29.8% (Fig. 4). For A. ostoyae, SNP in the repeat motif and indel in the flanking sequence also represent a significant proportion of the polymorphism detected (12.6% and 6.5% respectively). Quercus sp. were characterised by SNP both within the repeat motif (7.5%) and the flanking region (9.3%). The second most common polymorphism for Alosa sp. was SNP in the repeat motif (8.5%) and for M. variegatipes SNP in the flanking region (5.5%). For S. salar, variation in the number of motif was much more frequent than for other species (93.0%) compared to other polymorphisms that represent a marginal proportion of the variation (less than 3% each).

Table 3 Detected polymorphism.

						Polymorphism in the repeat motifc	Polymorphism in the flanking sequencesc	
Species	Number of loci	Number of alleles with sequence differencea	Number of alleles differing by amplicon sizeb	% of size homoplasy	% of increase in alleles due to sequence	Repeat number variation	SNP	Indel	SNP	Indel	
S. salard	14	122	108	11%	13%	107 (14)	3 (3)	–	2 (2)	3 (3)	
Quercus sp.	40	537	346	35%	55%	406 (40)	38 (25)	1 (1)	47 (18)	13 (10)	
Alosa sp.	24	174	150	14%	16%	130 (23)	13 (15)	2 (2)	4 (4)	3 (3)	
A. ostoyae	38	398	187	53%	113%	187 (33)	79 (26)	8 (7)	312 (23)	41 (16)	
M. variegatipes	39	166	156	6%	6%	147 (39)	3 (3)	1 (1)	9 (7)	5 (5)	
Notes.

a Irrespectively of polymorphism type, computed based on the Combined analysis strategy.

b Simulating the number of alleles that would have been identified using traditional capillary electrophoresis, computed based on the FullLength analysis strategy and accounting for allele size only on the same locus as the Combined approach.

c Total number of alleles (and number of loci in brackets) for each polymorphism type.

d Combination of two sequence based microsatellite genotyping protocols.

Figure 4 Proportion of detected polymorphism types within the repeat motif or in the flanking sequence for each sample per species group.

Discussion

While relying on previously developed microsatellite assays is far from optimal, we found that primer redesign around known locus or de novo microsatellite development based on strict criteria gives successful single highly multiplexed PCR amplification and sequencing for 20 to 40 loci. However, key initial factors need to be considered for efficient SSRseq setup.

Not all roads lead to Rome: navigating pitfalls when adopting HTS for microsatellite genotyping

Our preliminary attempts to apply SSRseq using previously developed microsatellite primers or capillary-based multiplexed microsatellite genotyping protocols clearly failed to produce reliable genotypic data for a sufficient number of loci. In the best case scenario, sequencing of 15 microsatellites in Salmo led to the reliable genotyping of 7 loci with 15.6% of missing data and 1.19% of allelic error, a result far worse than the high quality dataset obtained for the same multiplex using traditional capillary-electrophoresis of 14 loci with 0.5% missing data and 0.35% allelic error (Bacles et al., 2018). Similar trends were observed in previous studies that validated the use of HTS to genotype microsatellites and relied on previously developed primers: the generated datasets were characterised by high levels of missing data (up to 45%, Vartia et al., 2016) or low number of genotyped loci (7 to 8, Suez et al., 2016; Barbian et al., 2018). Two characteristics are problematic for SSRseq from microsatellites initially developed for capillary electrophoresis-based genotyping. First, high variability in locus length and primer characteristics leads to heterogeneous amplification intensity and sequencing coverage across loci. Indeed, efficient multiplexed PCR necessitates careful primer design using strict criteria (Guichoux et al., 2011; Lepais & Bacles, 2011). In addition, the need for variable locus length for optimal multiplexing without allele size range overlap in capillary-based electrophoresis becomes unnecessary for sequencing, because same size loci can be reliably identified simply based on primer sequences. Secondly, starting from a limited number of loci (e.g., 10-20 typical of capillary electrophoresis-based microsatellite genotyping approaches) results in a handful of reliable loci that may be too low for downstream applications. This conclusion agrees well with a previous study developing SSRseq in S. salar, where only one out of six (17%) of previously developed loci was successfully integrated into the final panel, compared with a 26% success rate for newly developed primers (Bradbury et al., 2018). Adapting previously developed loci in the same species resulted in a success rate of about 45% in our case (7 out of 15 and 10 out of 23 reliable loci), but de novo development in other species was much more successful with a success rate of 75% on average. We did not retrospectively apply the refined SSRseq development approach for S. salar to confirm it is also working on this species because it would duplicate recent protocol developed for this species (Bradbury et al., 2018) with limited usefulness for the scientific community. Nevertheless, our result clearly show that not relying of previous primer design is of primary importance for the success of reliable SSRseq protocol development.

Workflow for efficient SSRseq development in non-model species

We propose here a workflow for developing new SSRseq approaches and we demonstrated its efficiency for a range of species with different level of genomic resource availability. Starting from a reasonable number of candidate loci that were identified using readily available or newly generated genomic resources, resulted in an average of 75% of the loci included in the PCR multiplex generating reliable genotypic data. This result compares favourably with previous studies where success rates were lower (47% (Farrell et al., 2016), 53% (Neophytou et al., 2018)) or similar (78% (Tibihika et al., 2018)) when developing about 20 to 40 loci from a moderate number of candidate loci. However, extensive screening of numerous markers resulted in a much lower success rate: 101 validated loci from 385 tested (26% success rate (Bradbury et al., 2018)) or 43 validated loci from 448 locus tested (10% (Zhan et al., 2017)). Note however that leveraging extensive whole genome resequencing data can result in significant improvement of development success by targeting polymorphic microsatellites with perfect repeat motif and invariant flanking sequences (Yang et al., 2019). In this study, we propose a balanced approach for effective development that relies on simple amplification tests of a limited number of carefully selected candidate loci. Locus validation is then made when sequencing the final set of loci in a way that the validation step, based on the analyse of blind-repeat of at least 48 individuals, jointly generates useful genotypic data (for additional individuals included in the sequencing run, Table 2). In addition, most of the ordered primers will be validated and integrated in the final set of reliable loci which greatly minimise development costs. The workflow is also flexible in terms of number of loci chosen for analysis: if a higher number of loci is required; additional sets of 60 loci could be selected from the candidate locus list and amplified in separated PCR multiplexes that can be pooled together before PCR indexing leading to an effective way to genotype additional loci (Bradbury et al., 2018). We did not test multiplexing more than 60 loci in a single PCR, but we did not see difficulties in doing so (Campbell, Harmon & Narum, 2015). In such a situation, careful primer design with strict criteria and control for primer interactions will be key to increase multiplexed amplification success.

The proposed workflow for SSRseq development minimized laboratory steps and analytical optimizations. We chose to start from a moderate number of loci, designed to maximize their compatibility and sequence interpretability, and remove any locus that failed to amplify, produce interpretable or repeatable genotypes. This strategy avoids tedious optimization and reduce the number and complexity of laboratory steps necessary to produce genotypes. Admittedly, additional DNA or amplicon clean up or normalisation might improve sequence quality and coverage across individuals and loci. However, we chose to keep the laboratory procedure as simple as possible and compensate the increase in amplification heterogeneity by additional sequence output resulting in sufficient coverage (220 reads/loci/individuals on average) to obtain a nearly complete genotypic dataset (generally 5% of missing data excluding the atypical case of Armilaria species). Moreover, increasing the number of laboratory steps inflates the risk of handling error and subsequently of genotyping error (Vartia et al., 2016). In this respect, highly multiplexed PCR are a useful technique to reduce laboratory steps and potential associated errors in addition to saving time and cutting costs. Additional tests not presented here in addition to results from previous studies (De Barba et al., 2016; Zhan et al., 2017; Bradbury et al., 2018; Tibihika et al., 2018) showed that the two-stage multiplex amplification prior to PCR indexing that was used to increase amplification homogeneity across loci is not necessary: high coverage is still efficient to compensate the increase in amplification heterogeneity when using a single multiplex PCR step for locus amplification.

We chose to rely on the set of 384 Illumina barcodes combinations because we found it to fit well with the output of the MiSeq sequencing platform when analysing from 20 loci to 300 loci depending on the type of flow cell used. However, studying more than 384 individuals from a single species necessitates either several MiSeq runs (Bradbury et al., 2018) or custom made dual-indexing strategies, as was successfully performed for the Ion Torrent PGM run (960 barcode combinations used) or in previous studies using the MiSeq platform (960 and 1,024 barcode combinations (Farrell et al., 2016; Zhan et al., 2017)).

Finally, including repeated individuals is of paramount importance to assess the reliability of the produced genotypic data for each locus. This procedure aims to be best practice in capillary electrophoresis-based microsatellite genotyping (Hoffman & Amos, 2005; Pompanon et al., 2005; Guichoux et al., 2011) but is even more important in SSRseq. Indeed, not all sequenced loci produced reliable genotypic data. It is thus necessary to be able to identify and exclude loci producing high genotyping errors. At the bioinformatics analysis stage, selecting the best analytical strategy for each locus is an easy task that improve the number of reliable loci and decrease the genotyping error rate. However, a few loci will still show high genotyping error rate due to low coverage or complex polymorphism patterns and should be excluded from the final genotypic dataset. Exploratory analyses testing a much wider number of parameters resulted in limited success in increasing the final number of reliable locus. Extensive parameter testing is a tedious task requiring high computation time with only minor improvement for the final genotypic dataset quality and is not worth the extra effort as long as a sufficient number of loci have been included in the genotyping panel. Furthermore, additional analyses using alternative microsatellite sequences analysis tools such as Megasat (Zhan et al., 2017) and MicNeSs (Suez et al., 2016) also lead to the conclusion that all loci cannot be analysed reliably using a single set of parameters. Thus, we stress the importance to include a significant number of repeated individuals (at least 48) in the first analysis of a new SSRseq panel, irrespective of the bioinformatics data analysis strategy used, to (1) coarsely optimize analysis parameters for reliable loci, (2) quantify genotyping error rate and (3) identify and exclude unreliable loci. Such procedure has been implemented only in a limited number of previous studies describing SSRseq methods (6 out of 14 published studies thus far (De Barba et al., 2016; Zhan et al., 2017; Bradbury et al., 2018; Barbian et al., 2018; Šarhanová et al., 2018; Viruel et al., 2018)) but should be generalized. Downstream biologically-informed statistical analyses to verify that the loci comfort to Hardy-Weinberg equilibrium, detect the presence of null-alleles and estimate genotyping error rate based on sibship inference in natural population or known pedigree (Wang, 2017) should then be applied to further validate the obtained genotypes.

Implications of haplotype based genotyping

We took advantage of the fact that reads span across entire loci to analyse all linked and phased polymorphisms encountered using the FDSTools pipeline (Hoogenboom et al., 2016). This haplotype approach differs from the methods implemented in other sequence-based microsatellite genotyping software such as Megasat (Zhan et al., 2017) which focuses on amplicon length or Micness (Suez et al., 2016) which estimates the number of repeated motifs while accounting for up to one substitution within the microsatellites motif. The haplotype approach has several advantages. Firstly, it is relatively insensitive to sequencing error because the analysis focused on unique sequence with high coverage and thus does not consider the noise generated by sequencing error which produce numerous unique low coverage sequences that are removed. Secondly, by analysing unique sequences, it accounts for any kind of polymorphisms, while at the same time includes an algorithm to identify stutters resulting from slippage mutation due to microsatellite instability during PCR amplification.

Previous studies have demonstrated the added benefit of analysing different types of polymorphism within sequences compared to using amplicon size only (as in traditional capillary electrophoresis-based genotyping) to differentiate alleles (Darby et al., 2016; Neophytou et al., 2018; Barbian et al., 2018; Tibihika et al., 2018; Curto et al., 2019). Size homoplasy, due to alleles identical by size but not by sequence, ranges from to 32% and 64% (Darby et al., 2016; Vartia et al., 2016; Barbian et al., 2018; Šarhanová et al., 2018). Here, we found high variability in size homoplasy ranging from 6% to 53% between the studied species in direct correlation to the different types of variation observed across species. SNP in the repeat motif or in the flanking region was the main source of size homoplasy which showed high prevalence in A. ostoyae (SNP represented 62.4% of the detected variation and 53% of size homoplasy) and to a lesser extent in Quercus sp. (SNP represented 16.8% of the detected variation and 35% of apparent homoplasy). Even for species with lower apparent polymorphism levels, such as M. variegatipes (6%), S. salar (11%) and Alosa sp. (16%), the increase in the observed number of alleles will substantially improve genotypic resolution power (Darby et al., 2016; De Barba et al., 2016). Haplotype-based analysis that accounts for all linked variations across the whole sequence will make the most of the information available from sequence data (Barthe et al., 2012).

Finally, the ability to detect different sources of variation originating from several mutation mechanisms occurring at different rates provides renewed opportunities to study ecological and evolutionary events that occur at different timescales (Ramakrishnan & Mountain, 2004; Barthe et al., 2012). Combining information on linked microsatellites and SNP (into a system called SNPSTR (Mountain et al., 2002) or HapSTR (Hey et al., 2004; Sorenson & Dacosta, 2011)) was demonstrated to be a promising approach thanks to the increased phylogenetic resolution offered by explicitly considering complementary mutation properties of the markers. While theoretical and analytical implications of these approaches have been derived (Ramakrishnan & Mountain, 2004; Hey et al., 2004; Payseur & Cutter, 2006), empirical applications remain scarce and restricted to a very small number of systems due to the previous difficulties encountered to generate such empirical data (Mountain et al., 2002; Hey et al., 2004). This early limitation does not hold anymore with the generalisation of sequence-based microsatellite genotyping, as proposed herein, and the new ability to analyse linked microsatellites and SNP as haplotype. In addition, the flexibility of coalescent programs to simulate linked loci of different types (e.g., fastsimcoal2 (Excoffier et al., 2013)) will authorize far more realistic simulation of mutation mechanisms specifically tailored to each marker system. Such improvement, especially when applied to tens of loci, would make simulation-based inference (Beaumont, Zhang & Balding, 2002) more accurate over an extended timescale range even for complex evolutionary history scenarios (Mountain et al., 2002).

In conclusion, this study proposes an integrated approach to expedite the development of SSRseq protocol for non-model species and provides several recommendations to improve development efficiency. The two most important advices are to optimize marker selection and primer design for effective multiplex PCR amplification and sequence interpretability, and to use repeated individuals to assess the quality of the generated genotypic data. The ability of SSRseq to characterize SNP and indel present along the sequences, in addition to the targeted microsatellite, represents a new opportunity to produce empirical data to apply existing theoretical and statistical frameworks that integrate linked polymorphism with different mutation characteristics (Payseur & Cutter, 2006). Genotyping relying on sequence data that are easier to normalize than traditional capillary electrophoresis genotyping through automated bioinformatics pipelines will facilitate sharing of data between laboratories and incrementing genotypic database that are paramount for applications in wildlife monitoring (Bradbury et al., 2018; Layton et al., 2020) or agronomical research (Li et al., 2017; Yang et al., 2019). Finally, the ease of parallel development for multiple species make these approach convenient to develop powerful multilocus datasets for comparative population and community genetics studies (Crutsinger, 2016), and to further investigate the functional implications (Bagshaw, 2017) and adaptive potential of microsatellite variation among natural populations (Xie et al., 2019).

Supplemental Information

Supplemental Information 1 Detailed description of FDSTools parameter sets used (PS1 and PS2)

Click here for additional data file.

Figure S1 Effect of sequence coverage (using one, two or three Illumina Miseq nano flowcells) on SSRseq data quality for Alosa species

Click here for additional data file.

Table S1 Detailed characteristics of all loci tested in this study

Characteristics of all tested loci in this study, including primer sequences, reference, repeat motif size, success of amplification and genotyping, bioinformatics pipeline parameter, number of haplotype and alleles based on size, missing data rate and allelic error rate, type of polymorphisms detected.

Click here for additional data file.

Technical developments and sequencing were performed at the Genome Transcriptome Facility of Bordeaux. We thank François Meurgey for the sampling of Melipona sp. individuals.

Additional Information and Declarations

Competing Interests

Author Contributions

Data Availability

The authors declare there are no competing interests.

Olivier Lepais conceived and designed the experiments, performed the experiments, analyzed the data, prepared figures and/or tables, authored or reviewed drafts of the paper, and approved the final draft.

Emilie Chancerel and Laura Taillebois performed the experiments, analyzed the data, authored or reviewed drafts of the paper, and approved the final draft.

Christophe Boury, Franck Salin and Erwan Guichoux conceived and designed the experiments, performed the experiments, analyzed the data, authored or reviewed drafts of the paper, and approved the final draft.

Aurélie Manicki, Abdeldjalil Aissi and Cecile F.E. Bacles performed the experiments, authored or reviewed drafts of the paper, and approved the final draft.

Cyril Dutech, Françoise Daverat and Sophie Launey conceived and designed the experiments, authored or reviewed drafts of the paper, and approved the final draft.

The following information was supplied regarding data availability:

The sequence reads from SSRseq run are available at the European Nucleotide Archive SRA: PRJEB31908 (Alosa sp.), PRJEB31909 (A. ostoyae), PRJEB31910 (M. variegatipes), PRJEB31913 (Quercus sp.) and PRJEB31914 (S. salar) and the random shotgun sequencing reads from one M. variegatipes individual for microsatellite discovery are available at: ERR3255838.

Pipeline scripts for automated sequence-based microsatellite genotyping with documentation and example files are available at Data Inra: Lepais, Olivier, 2019, “Automated pipeline for sequence-based microsatellite genotyping”, https://doi.org/10.15454/HBXKVA, Portail Data INRAE, V2.

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
