# Peer review of "Fast sequence-based microsatellite genotyping development workflow"

_PeerJ, doi:10.7717/peerj.9085_

## Round 0.1 · original submission · Major Revisions

Both reviewers have provided thorough comments on the manuscript, and these can be found below.

Reviewer 1 ·

Basic reporting

The study is well-written, although some smaller errors are highlighted in point 8 under specific comments. I suggest either shortening or rewriting the introduction (move around some sections of the intro), for example starting with how was it done before (Line 62) why is SSR important and has gain more attentions lately (line 48-61) and how should we use this markers now in light of new technologies, see point 4 & 6 under General/specific comments.

Raw data is available as stated by the authors. Figures and table are easy to understand, but see point 7 below.

Supplementary data (scripts, fastq, genotyping details etc) is accessible and followed by good documentation for executing the necessary analysis steps.

Experimental design

The study is well suited to the scope of this journal.

The research question concern improved quality of genotyping and broader use of a well-known marker system coupled to new sequencing technologies as well as the implementation of an efficient marker selection procedure/workflow. This is well described and sufficiently detailed for replication. Additional test/experiments to prove the concept is still valid see point 1 & 2 below.

Validity of the findings

Validity of the findings are clear, but I would like to see the suggested test on salmon performed, as well as a better discussion of point 4 and 5 below. Besides these comments, the conclusion is valid.

Additional comments

The authors provide a nice overview on how quickly and reliably one can detect SSRs across phyla, improve genotyping success, ultimately leading to better and more robust analyses of evolutionary processes. It’s a nice study with the aim on integrating many protocol steps, so its of no surprise that many of the steps has been described and used before (or similar/related development and analysis steps). Similar methods is also available from commercial suppliers (https://www.cd-genomics.com/hi-ssrseq.html). Still, I feel it`s important to make such integrative approaches as it makes this important tool more easily accessible to new users, particular users unfamiliar with NGS technologies. In that respects the manuscript provide an easy and quite simple entry into better development of SSR-seq markers. I also like that development tools has both command line and GUI interface with Galaxy for users with different skill levels and preferences.

Specific comments
Below I´ll list in order of importance different subjects that hopefully could improve the manuscript.

1. It would be nice to see the application of your SSRseq method on the Salmon as well. After all this is the only species you have compared the efficiency of capillary electrophoresis (CE) based loci with direct sequencing. It seems a bit premature to focus on other species until these new SSR-seq markers for salmon are developed and tested as the other species. Furthermore, it will make the comparison with other work on salmon, which your refer to in both result and discussion more interesting (Bradbury et al 2018.)

2. Next, since its salmon, and as close to a natural model organism you can get, a comparison of two different sets of newly developed SSR-seq loci (20-40 in each set) used on either samples of well-known natural populations or several farmed salmon families with known pedigree is valid. This will add strength to the suggested primer development and selection criteria you suggest, as genetic differentiation and/or family histories/relationships should be similar using either set of loci.

While both 1. and 2. above may seem unnecessary, it will strengthen the overall quality of the study. While CE msats has been difficult to a use over larger geographic scale, as well as being difficult to use with similar sensitivity and accuracy between labs, which is of importance for management purposes (e.g. fisheries and large carnivore management), the inclusion of the suggested tests might be convincing evidence for adopting this step-by-step procedure for monitoring wildlife.

3. Besides whole genome sequencing reads, i.e. short reads archive, low coverage genomes to more or less fully annotated genomes, please consider if the MSDB (microsatellite database https://data.ccmb.res.in/msdb/, https://www.ncbi.nlm.nih.gov/pubmed/31599331 ), or similar single species databases (crop species primarily) could be a valuable source for finding repeat loci.

4. RAD-seq is mentioned briefly in the intro. By many NOT considered expensive, in contrast to the authors claim. Besides its flexible, same RE and reagents used on different species. Other new marker types like MobiSeq (https://doi.org/10.1111/1755-0998.12984) and other reduced representation seq data, as well as their cost, should be mentioned and discussed. After all SNPs do have a simpler mode of evolution. What does SSRseq offer that is better? (its touched upon but higher level of polymorphism could be made better selling points if properly backed up by other studies).

5. On the subject of primer and loci optimization, Yang et al 2019 (https://doi.org/10.3389/fpls.2019.00531) used “perfect SSR loci”, or according to them “stable motifs and flanking sequences”. They aimed for higher sequence coverage, and had a remarkable high genotyping success, which should be commented on.

6. Introduction. Sentence 31-33. Maybe skip the general NGS overview. Not including medicine for instance in fields using HTS is questionable. So, include more research fields or go directly onto SSR-seq (as done in abstract). A better structure could be to start introduction from Line62, the old way of doing it. Then, line 48-62, which highlight new biological aspects of SSR, followed by a new way of generating robust and reliable SSR markers by sequencing.

7. Figure 2 & 3. Please name colors in parenthesis in the text for each parameters.
8. Some typos and suggestions (your call), numbers refer to line/sentence no
a. 41: SNPs
b. 62: include “traditional” in front of msat
c. 98: with instead of which
d. 103: is this really a “new” marker?
e. 123: to avoid confusion with sequencers just use “standard capillary electrophoresis (CE)”
f. 191: use “simplex amplification one individual per species”
g. 372: much higher
h. 382: assays
i. 414: resource

Reviewer 2 ·

Basic reporting

no comment

Experimental design

no comment

Validity of the findings

no comment

Additional comments

This is an interesting and well-written paper that describes a workflow to efficiently identify microsatellite markers and genotype them using SSRseq. The authors' findings suggest the importance of sequencing coverage and duplicated individuals in the experimental design, in order to reduce missing data and eliminate markers that cannot be genotyped reliably.

Major points:
282-284 “Loci that showed more than 6% of allelic error or more than 50% of missing data across individuals (within each species group) were flagged as failed and removed from further inspection.”

Could you explain how these thresholds were established? 50% of missing data seems to be a very high level. What happens if the threshold for missing data and allelic error is decreased? Would this eliminate a high proportion of loci? Figures 2 and 3 suggest that the overall missing data and allelic error rate are not so high, but we do not have the information on the distribution of these indices among loci. It would be important to calculate missing data and allelic error rate for each locus separately, and represent the variability of these indices on Figures 2 and 3.


339- 342 “With the pipeline parameters used here, increasing coverage will reduce missing data but not genotyping error rate, however with more stringent pipeline parameters, increasing coverage may have more significant impact on the number of reliable loci genotyped and missing data rate.”

Since you are suggesting anyway that a relatively high coverage is important to reduce missing data, it would be interesting to test if more stringent parameters in allele calling would really reduce allelic error rates in high coverage data. That would be a supplementary argument for increasing data coverage.

344-345 “Focusing the analysis on the repeated motif slightly increased the number of reliable loci and tends to produce fewer missing data and allelic errors (Figure 3).”

The change in missing data and allelic errors does not seem to be evident from figure 3. Please, modify this statement.


447-449 “Additional tests not presented here showed that the two-stage multiplex amplification prior to PCR indexing that was used to increase amplification homogeneity across loci is not necessary:”

It would be nice to present this finding here, since it simplifies even further the laboratory protocols.



Typos and minor suggestions:

Line 252 ‘FInallt’

257 …’genotypes divided by the number of diploid alleles comparred’ => An allele cannot be diploide; comparred=> compared

268 ‘FullLenghth’

363- 365 Conversely… (Table 3).
This sentence is redundant to the previous one. Similarly, the columns ‘% of size Homoplasy’ and ‘% of increase in alleles due to sequence’ in Table 3 are redundant.

372 ‘High’ => higher

---

## Round 0.2 · accepted · Accept

Both reviewers were happy with the revisions made to your manuscript (bar one typo), so are happy to accept. Congratulations.

Reviewer 1 ·

Basic reporting

The same general impression of a clearly written paper, as from the first round of review.
The structure is now much better and more directly to the point, which is most important for such a detailed protocol paper.

Improvement to figures are good, but see typo "usuing" in figure text for "Supporting figure S1".

Experimental design

The additional comments, se review round 1

Validity of the findings

The additional comments, se review round 1

Additional comments

The authors has largely followed both reviewers suggestions, and/or provided good explanations for why they choose otherwise. That perfectly fine. The structure has improved, and its more interesting to read, down to the points that matters. I thank the authors for presenting this study. Its a clear improvement from the CE based method, and support the community with a step by step method suited for comparison between labs and regions, enabling cheap and robust genotyping over the whole distrubution ranges of species.

Reviewer 2 ·

Basic reporting

no comment

Experimental design

no comment

Validity of the findings

no comment

Additional comments

The authors have answered convincingly all the points I raised in my first review.